# Quality and Pro-Healthy Properties of Belgian Witbier-Style Beers Relative to the Cultivar of Winter Wheat and Raw Materials Used

**DOI:** 10.3390/foods11081150

**Published:** 2022-04-15

**Authors:** Justyna Belcar, Jan Buczek, Ireneusz Kapusta, Józef Gorzelany

**Affiliations:** 1Department of Food and Agriculture Production Engineering, Collegium of Natural Sciences, University of Rzeszów, 4 Zelwerowicza Street, 35-601 Rzeszow, Poland; gorzelan@ur.edu.pl; 2Farming Cooperative SAN, Łąka 598, 36-004 Łąka, Poland; 3Department of Crop Production, Collegium of Natural Sciences, University of Rzeszów, 4 Zelwerowicza Street, 35-601 Rzeszów, Poland; jbuczek@ur.edu.pl; 4Department of Food Technology and Human Nutrition, Collegium of Natural Sciences, University of Rzeszów, 4 Zelwerowicza Street, 35-601 Rzeszow, Poland; ikapusta@ur.edu.pl

**Keywords:** wheat, wheat malt, Witbier-style beers, beer quality, polyphenols, antioxidant potential of beer, sensory evaluation

## Abstract

Unmalted wheat grain and barley malt are the basic materials used in the production of Belgian wheat beers known as Witbier. A change in the ingredients defined in the recipe, by which part of the unmalted wheat is replaced with wheat malt, can positively affect the quality of the beverage produced. The purpose of the study was to brew Witbier-style beers made from four cultivars of winter wheat, with a 50% share of unmalted wheat and barley malt as well as Witbier-style beers made from four wheat cultivars, where 25% of unmalted wheat was replaced with wheat malt. Physicochemical and sensory analyzes showed mild differences in the quality of the beer products, more specifically higher alcohol content (by 11.33%) were found in beers made without the addition of wheat malt, while higher sensory attractiveness and 17.13% higher total polyphenol content were identified in beers enhanced with wheat malt. Phenolic compounds were identified using UPLC-PDA-MS/MS. The highest flavanol content, including kaempferol 3-O-rhamnoside-7-O-pentoside, was found in beers produced using wheat grains of the ‘Elixer’ cultivar, whether or not wheat malt was added; the values were 1.31 mg/L in E50 beer, and 1.39 mg/L in E25 beer. The same beer samples with the highest antioxidant and antiradical activity were found (in E25 beer, 2.35 mmol TE/L, and in E50 beer, 2.12 mmol Fe^2+/^L). The present findings show that the investigated wheat cultivars may be used in beer production, whereas replacing part of unmalted wheat with wheat malt can improve the sensory profile of the beer produced.

## 1. Introduction

In recent years, wheat beers have become very popular, not only in Poland but also worldwide. Ingredients of barley and wheat beers include malt, hops, yeast, and water. In relation to barley beers, wheat beers differ with the raw material used, that is, part of barley malt is replaced with wheat malt or unmalted wheat (most commonly at a rate of 40 to 60% of the total raw material) [1]. Wheat beers are products of top fermentation and, depending on the type, they are characterised by rich flavour and aroma (e.g., malty, bready, wheaty, with notes of clove, vanilla, banana, or other, resulting from the yeast strain selected for fermentation), light and stable frothy foam, haziness and slightly bitter taste [2,3,4]. The degree of haziness of wheat beer depends on interactions between (gluten) proteins and starch or polyphenols [5]. The stability of the foam of the beer is determined by the addition of adequate amounts of hops and by the contents of ions of metals in the wort, whereas the limiting factors include lipids and a high concentration of alcohol in wheat beer [3].

Although they contain ethanol, wheat beers also have high contents of compounds known for their bioactive and antioxidant effects and for their nutritional value, such as vitamins, minerals, amino acids, and polyphenols [6,7]. The latter, being the most important group of antioxidants, affect certain sensory qualities of beer, e.g., pungency, haziness and depth of taste [8,9]. Antioxidants protect the human body against oxidative stress; however, they are highly sensitive to temperature, pH, oxygen level, light, and yeast. Stress factors occurring during fermentation, e.g., too high temperature, too high concentration of ethanol, deficiency of nutrients for yeast, have a negative impact on yeast cells, which, defending themselves, produce reactive oxygen species damaging their structure. Reactive compounds pass to fermenting wort, in which they combine, among others, with elements such as iron or copper and form hydroxyl radicals, which contribute to the damage of compounds with antioxidant activity affecting the taste of beer. During the beer aging process, they are responsible for changes in flavour [10]. Through their activity, antioxidants neutralise free radicals that attack and damage the structure of DNA, membrane lipids, and proteins [11].

Belgian wheat beers, known as Witbier, are made from raw materials which include unmalted wheat and pale barley malt (Pilzner type), as well as characteristic flavour enhancers, i.e., curaçao bitter orange peel and coriander. These are light and cloudy beers with pleasant aroma and characteristic citrus flavour. Witbier is characterised by a stable and strong beer head and colour ranging from pale yellow to golden yellow, and having low content of bitter substances. A change in the ingredients defined in the recipe for Witbier-style beer, whereby part or all of the unmalted wheat is replaced with wheat malt, affects the sensory properties of the beer, including its flavour and colour [12,13]. Using unmalted grain in the production of wheat beers, it is also possible to reduce the production costs associated with the malting and drying processes, which can also be achieved due to the availability and lower price of unmalted raw material, for example, wheat of high technological quality [14].

The purpose of the study was to compare sensory properties, physicochemical characteristics, polyphenol profiles, and antioxidant activity in Witbier-style beers, relative to the winter wheat and ingredients used. It also aimed to determine the potential of winter wheat cultivars as raw material to be used in beer production.

## 2. Materials and Methods

### 2.1. Material

A standard Witbier recipe was applied to produce eight wheat beer samples. Four winter wheat were applied. The ‘Gimantis’ and ‘Elixer’ wheat grains were produced in a field experiment conducted in 2021 in the village of Kosina (50°04′17″ N 22°19′46″ E), Podkarpackie Region (Poland); The ‘Rockefeller’ wheat grain was obtained in a field experiment conducted in 2021 in Głuchów (50°04′54″ N 22°16′11″ E), Podkarpackie Region (Poland), and the ‘Lawina’ wheat grain was produced in a field experiment conducted in 2021 in Lipnik (49°59′31″ N 22°18′52″ E), Podkarpackie Region (Poland). The grain of winter wheat cultivars was harvested after achieving full maturity, and after a resting period, it was used to prepare five-day wheat malts (the methodology of the malting process was described by Belcar et al. [15]).

The materials used in the production of the beer samples included commercially available barley malt acquired from the Viking Malt company in Strzegom (Poland). Wheat and barley malts, as well as unmalted grains of the investigated wheat cultivars, were refined to the required particle size using a Cemotec disc mill manufactured by FOSS (Sweden). The beer samples were divided into two groups, based on the varied proportions of the raw materials used.
−Input materials with 50% share of unmalted wheat grains and 50% of Pilzner type barley malt: beer sample made from ‘Gimantis’ wheat grain was marked as G50 (parameters of wheat grain: protein content—14.07% d.w.; starch content—59.6% d.w.); beer sample made from ‘Elixer’ wheat grain was marked as E50 (parameters of wheat grain: protein content—12.71% d.w.; starch content—61.9% d.w.); beer sample made from ‘Rockefeller’ wheat grain was marked R50 (parameters of wheat grain: protein content—11.14% d.w.; starch content—63.2% d.w.); beer sample made from ‘Lawina’ wheat grain was marked L50 (parameters of wheat grain: protein content—12.93% d.w.; starch content—61.7% d.w.);−Input materials with 25% share of unmalted wheat grains, 25% of Pilzner type wheat malt and 50% of barley malt: beer sample made from wheat grain and wheat malt of ‘Gimantis’ cultivar was marked G25 (parameters of wheat malt: protein content—13.67% d.w.; extractivity—87.5% d.w.; diastatic power—318 WK); beer sample made from wheat grain and wheat malt of ‘Elixer’ cultivar was marked as E25 (parameters of wheat malt: protein content—11.83% d.w.; extractivity—86.4% d.w.; diastatic power—329 WK); beer sample made from wheat grain and wheat malt of ‘Rockefeller’ cultivar was marked as R25 (parameters of wheat malt: protein content—10.77% d.w.; extractivity—85.6% d.w.; diastatic power—303 WK); beer sample made from wheat grain and wheat malt of ‘Lawina’ cultivar was marked L25 (parameters of wheat malt: protein content—12.71% d.w.; extractivity—84.9% d.w.; diastatic power—320 WK).

### 2.2. Production of Beer

The Witbier samples were produced in a laboratory at the Department of Agricultural and Food Engineering at the University of Rzeszów. Each beer sample was made from 5.0 kg of raw materials. The refined raw materials were placed in the ROYAL RCBM-40N brew kettle (Expondo; Zielona Góra; Poland; applied with 80% process efficiency), with 15 L of water (3 L of water per each kilogram of the raw material). The mashing process was divided into the following stages: 5 min at a temperature of 52 °C, 60 min at 67 °C, 20 min at 72 °C, and 10 min at 78 °C. After an iodine starch test produced a negative result, the mashing process ended, and the mash was subjected to filtration and sparing with water at 78 ℃.

Sweet wort was placed in the brew kettle ROYAL RCBM-40N and heated up to a temperature of 100 °C, at a rate of 2 °C/1 min. The wort was boiled for 60 min and during this time Lubelski variety hops were added (Poland; α-acid contents of 5.7%) in two doses: when the wort reached the boiling temperature (at 0 min), 20 g of hops, and then after 45 min of boiling, 10 g of hops. In the 55th minute during the boiling process, the following ingredients were added: 15 g of coriander grains, 15 g of curaçao orange peel and 1 g of chamomile herb.

Subsequently, the hot wort was cooled using a spiral immersion cooler and tap water as a cooling agent. After 30 min, the wort decreased to 20 °C. The eight wort samples had extract content of 11.0 °P. The cooled wort was poured into 30 L fermentation vessels. Subsequently, the yeast *Saccharomyces cerevisae* Fermentis Safale S-33 (6 × 10^9^/g), earlier subjected to a dehydration process (0.58 g d.m./1L of wort), was added to the vessels. The fermentation process was carried out at 21 °C. After the 14-day fermentation process, an aqueous solution of sucrose (0.3%) was added, and the beer was poured into bottles for refermentation to achieve adequate saturation. The beer samples were then stored at a temperature of 20 °C. Sensory evaluation, antioxidant activity evaluation, and physicochemical tests were performed one month after bottling.

### 2.3. Analysis of the Quality Characteristics of the Wheat Beers Produced

The characteristics of the fermentation process were determined according to method 9.4 [16], by calculating the following three values: apparent extract (AE) measured as the specific weight of beer at 20 °C, after alcohol has been distilled from the beer; original extract (OE) measured as the specific weight of the wort at 20 °C, before fermentation, as well as real extract (RE) measured as the specific weight of beer at 20 °C after fermentation has been completed. Parameters that reflect the activity of yeast during fermentation, that is, apparent attenuation (AA) and real attenuation (RA), were determined using the following formulas [2,17]:AA = (OE − AE)/OE × 100(1)
RA = (OE − RE)/OE × 100(2)

The beer samples were examined for the carbon dioxide (PN-A-79093-6:2000) [18], bitter substances, according to the method 9.8 (International Bitterness Units—IBU) of the EBC [19], and alcohol—in accordance with EBC method 9.2.3 of the EBC [20]—whereas pH values were determined according to the method 9.35 [21]. The colour of the beer samples was measured in units of EBC, according to EBC method 9.6 [22]. The titratable acidity of wheat beers was determined by subjecting beer samples to titration with 0.1 M NaOH, with end point at pH = 8.2 [23]. The energy value of wheat beers was calculated following the formula: [kcal/100 mL] = (7 × A (% *v*/*v*) + (4 × Er (% *v*/*v*) × ρ) [24].

### 2.4. Contents of Bioactive Compounds in Witbier-Style Beers

#### 2.4.1. Antioxidant Activity of Beer Samples

##### DPPH Test

The antiradical activity of wheat beer was determined using the DPPH radical, according to the method proposed by He et al. [10]. For this purpose, a 0.05 mmol/L solution of DPPH (2,2-diphenyl-1-picrylhydrazyl) in ethanol was prepared. A 7.8 mL sample of the solution was placed in a test tube, with 0.2 mL of diluted (2x) beer and incubated in darkness for 60 min at a temperature of 37 °C; subsequently, the absorbance at wavelength λ = 517 nm was examined using a UV-Vis V-5000 spectrophotometer (Shanghai Metash Instruments Co. Ltd., Shanghai; China). The blank contained distilled water instead of beer. The results were expressed as Trolox equivalent (mmol TE/L). The analyses were performed in three replications.

##### FRAP Test

The reduction power of wheat beers was determined using the FRAP reagent, according to the method described by Benzie and Strain [25] and He et al. [10]. Materials prepared for this purpose included a 10 mmol/L TPTZ solution (2,4,6-tripyridyl-s-triazine), a 20 mmol/L FeCl_3_^.^6H_2_O solution, an acetate buffer with pH = 3.6, as well as a 40 mmol/L HCl solution. Subsequently, the FRAP reagent was prepared by mixing 25 mL of the acetate buffer with 2.5 mL of the TPTZ dissolved in HCl and 2.5 mL of FeCl_3_^.^6H_2_O. A 6 mL sample of FRAP solution was placed in a test tube with 0.2 mL of beer and incubated at a temperature of 37 °C for 10 min; subsequently, the absorbance at the wavelength of λ= 593 nm was examined using a UV-Vis V-5000 spectrophotometer (Shanghai Metash Instruments Co. Ltd., Shanghai; China). The blank contained distilled water instead of beer. The results of the FRAP test were expressed as mmol Fe^2+^/L. The analyses were performed in three replications.

#### 2.4.2. Total Contents of Polyphenols

The total polyphenol content [mg/L] in the investigated beer samples was determined using spectrophotometric method described by Dvořáková et al. [26], in accordance with EBC method 9.11 [27]. The analyses were performed in three replications.

#### 2.4.3. Determination of Polyphenols Profile by UPLC-PDA-MS/MS

Determination of polyphenolic compounds was carried out using the UPLC equipped with a binary pump, column and sample manager, photodiode array detector (PDA), tandem quadrupole mass spectrometer (TQD) with electrospray ionization (ESI) source working in negative mode (Waters, Milford, MA, USA) according to the method of Żurek et al. [28]. Separation was performed using the UPLC BEH C18 column (1.7 µm, 100 mm × 2.1 mm, Waters; Warsaw, Poland) at 50 °C, at flow rate of 0.35 mL/min. The injection volume of the samples was 5 µL. The mobile phase consisted of water (solvent A) and 40% acetonitrile in water, *v*/*v* (solvent B). The following parameters were used: capillary voltage of 3500 V; con voltage of 30 V; con gas flow 100 L/h; source temperature 120 °C; desolvation temperature 350 °C; and desolvation gas flow rate of 800 L/h. Polyphenolic identification and quantitative analyses were performed on the basis of the mass-to-charge ratio, retention time, specific PDA spectra, fragment ions and comparison of data obtained with commercial standards and literature findings. The analyses were performed in three replications.

### 2.5. Sensory Analysis in Beers

Sensory analysis of Witbier-style wheat beers was performed using a 5-point scale, assessing the specific quality characteristics, i.e., aroma (5—very strong, distinctive and pleasant; 1—imperceptible/ unpleasant smell), taste (5—very good; 1—bad); beer foam (5—highly stable; 1—unstable), bitter taste (5—weak; 1—very strong) and carbonation (5—high; 1—poor or none). The average score described the general impression (5—excellent; 1—poor) related to the wheat beers. A 5-point scale was also applied in the assessment of the quality of the beer foam: colour (5—white; 1—brown), abundance (5—highly abundant; 1—none), structure (5—fine and bubbly; 1—loose), and stability (5—highly stable; 1—disappears quickly). The average score described the general impression (5—excellent; 1—poor) related to the quality of the beer foam. The assessment team consisted of 15 trained appraisers. Coded samples of beer were placed in 250 mL plastic cups were given to the assessors in random order.

### 2.6. Statistical Analysis

The statistical analyses of the results were computed using Statistica 13.3 (TIBCO Software Inc., Tulsa, OK, USA). Results related to physicochemical characteristics, polyphenol content, antioxidant activity and sensory evaluation of Witbier-style wheat beer samples were examined using completely randomized ANOVA, with significance level α = 0.05. The mean values were compared using the Tukey HSD test.

## 3. Results and Discussion

### 3.1. Physicochemical Characteristics of Witbier-Style Beers

The raw materials used in the production of beer include mainly cereals with high contents of starch that is hydrolysed to fermentable sugars by amylolytic enzymes activated and synthesised during the malting process [29]. If unmalted grains, e.g., wheat, are added to the raw materials used, the mashing process, and such addition may affect the taste of the finished product [30].

The results showing the physicochemical parameters of wheat beers made from four common wheat cultivars and enhanced with wheat malt can be seen in Table 1.

The contents of apparent extract, real extract and original extract were significantly varied; the highest values were identified in the L50 sample. Compared to samples produced with 50% share of unmalted wheat grain, beers with wheat malt added were found with lower contents of apparent extract, on average by 13.02% m/m, real extract, on average 13.36% m/m, and original extract, on average by 11.65% m/m. Beers produced from unmalted wheat grain of ‘Elixer’ cultivar (E50) or with addition of wheat malt (E25) were found with the smallest differences in the original extract values, while the lowest differences in apparent and real extract contents were identified in beers made from ‘Gimantis’ wheat (G50 and G25) with values of 4.06% and 5.52%, respectively (Table 1). The recipes for Witbier-style beers specify that the original extract should be in the range of 11–13 °P, whereas the contents of final apparent extract should be 2–3 °P [12]. The apparent extract defines the concentration of soluble compounds left in the beer after fermentation [12]. All the investigated beer samples had apparent extract that was higher than the standard value; moreover, the beer samples enhanced with wheat malt were found with lower values of this parameter, which may be related to greater availability for yeast of the chemical compounds contained in the wort. Unmalted wheat grain, added to the raw materials, impacts effectiveness of fermentation process, as well as chemical composition of wort, alcohol content and real extract content, and consequently, it also affects the quality of the final product. This is associated with the lack of transformations resulting from cytolytic, proteolytic, and amylolytic processes in the grain, as it has not been subjected to the malting process [30,31].

The highest degree final apparent attenuation and degree of final real attenuation was identified in E25 sample (70.07% and 67.86%, respectively; Table 1). The value of these parameters was varied, and relatively low for the cultivars of unmalted wheat and wheat malt added to the raw materials. The low degree of final real attenuation in Witbier-style beers resulted in lower alcohol content. Belgian-type Witbier should have an alcohol content in the range of 4.5–5.5% *v*/*v* [12,13]. Higher alcohol contents were found in beer samples made from unmalted wheat and barley malt (1:1), compared to samples made with addition of wheat malt (Table 1). On the other hand, Costa et al. [12] produced Witbier-style ginger beer and reported alcohol contents ranging from 6.5 to 6.7% *v*/*v*. For comparison, de Freitas et al. [13] identified alcohol content of 4.1% *v*/*v* in beer made from unmalted wheat grain and 4.7% in beer made from wheat malt added at a rate of 50%, while in beers made with 30% and 60% share of unmalted wheat alcohol, the contents amounted to 3.7% *v*/*v* and 4.1% *v*/*v*, respectively [30]. In a study by Vogel [32], Witbier-style beer had 4.7% *v*/*v* content of alcohol. The lower content of ethyl alcohol resulted in lower energy value of wheat beers, ranging from 40.30 to 42.30 kcal/100 mL in the case of beers produced with the addition of wheat malt and 43.94 to 49.79 kcal/100 mL in the samples produced with 50% share of unmalted wheat grain (Table 1).

The colour of beers made from unmalted wheat and barley malt was significantly lighter compared to the colour of beers produced with wheat malt added; this is associated with the high content of high-molecular-soluble protein compounds in beers produced from unmalted wheat that were not subjected to proteolysis during a malting process. These compounds also contribute to the hazing of beer, because during storage they fall to the bottom of the bottle; on the other hand, beers enhanced with wheat malt contain low-molecular protein compounds (protein-polyphenol complexes) that do not tend to the settle, and contribute to clouding of beer [5]. Witbier-style beers are not subjected to filtration, which also contributes to a sense of increased clouding [13]. The grain that is not subjected to malting and then drying has a lower content of Maillard reaction products, e.g., melanoidins, which pass into the beer, producing its darker colour [2,17,30]. Furthermore, proteins that are passed into the wort during the mash and boiling with hops react with polyphenols, making it possible for sediments to form in beer, also affecting the colour [33].

The bitterness of beer is determined mainly by iso-α-acids and bitter acids derived from hops. The variety and quality of hops, as well as the boiling time, affect the content of bitter substances in the produced [33]. The latter value in all the assessed samples complied with the standards, according to which the contents of bitter substances in Belgian-type Witbier should be in the range of 10–20 IBU.

The carbon dioxide in the investigated beer samples investigated was in the range of 0.43–0.47% (Table 1). A study by Mascia et al. [17], investigating various types of wheat beer, reported the contents of carbon dioxide content ranging from 0.49% to 0.74%, while the values observed by Depraetere et al. [5] were in the range of 0.57–0.60%. The optimal carbon dioxide results in carbonation of wheat beer and contributes to the refreshing effect produced by beer; beers with low level of carbonation are not accepted by consumers.

The acidity of the Witbier-style beers was significantly varied and ranged from 2.04 in the R25 sample to 2.92 mL (0.1 M NaOH/100 mL) in L50 sample. Significantly higher acidity was found in beer samples produced with addition of wheat malt (Table 1). De Freitas et al. [13] obtained the acidity parameter in Witbier samples at a level of 2.29–2.40 mL (0.1 M NaOH/100 mL), depending on the composition of the raw materials. The acidity of beers is affected by the quality of the raw materials used and the biological activity of yeast. The higher acidity of beer is associated with a higher microbiological risk or with a lower effectiveness of the yeast strain during the fermentation process [13]. The pH value in wheat beers made from raw materials comprising wheat malt was lower, on average by 8.42% compared to beer samples made from raw material with 50% share of unmalted wheat (Table 1). Similarly, de Freitas et al. [13] reported pH values of 4.48 and 4.32 measured in Witbier-style beers from unmalted wheat and wheat malt, respectively; meanwhile, Yorke et al. [30] investigated barley beers made with a 30% or 60% addition of unmalted wheat and found pH values of 4.38 and 4.41 in the respective samples. A slightly higher pH value in the Witbier sample reported by Vogel [32] was 4.70. A low pH value is associated with a high content of free hydrogen ions in a beer product; these contribute to an increased sense of acidity [30,34]. A lower pH value reduces the development of undesirable microflora, favourably affecting the microbiological stability of beer [13,24].

### 3.2. Contents of Bioactive Compounds in Witbier-Style Beers

The antioxidant substances (mainly polyphenols, as well as vitamins, fibre, and bitter acids) contained in beer are derived mainly from malt and hops [33,35]. According to the recipe for the investigated Witbier-style beers, the same dose of freeze-dried hops was used in the production of all samples, hence the differences in the contents of bioactive compounds are mainly related to the wheat cultivar applied and the use of wheat malt as one of the raw materials.

The antioxidant activity of wheat beer samples was determined using DPPH and FRAP methods. The highest values were identified in beer samples made from ‘Elixer’ wheat (E50), with wheat malt added (E25) and samples made from ‘Rockefeller’ wheat (R50). The enhanced with wheat malt beer was found with antioxidant activity that, on average, was 20.23% lower compared to the wheat beer samples made with a 50% share of unmalted wheat, with the exception of E25 sample (Table 2). The increased activity of the DPPH radical in wheat beer reflects decrease in contents of aldehydes adversely affecting stability of taste, e.g., trans-2-nonenal [10]. However, the reducing power of Witbier-style beers, assessed using the FRAP method, was varied in the samples made of unmalted wheat and those enhanced with wheat malt, although the differences were not always significant (Table 2). The highest value was identified in the E50 sample—2.12 mmol Fe^2+^/L—and the lowest in the R25 sample—1.71 mmol Fe^2+^/L. A study by He et al. [10] showed that the reducing power of unpasteurized wheat beer was in the range of 1.48–1.71 mmol Fe^2+^/L. Technological processes that allow one to achieve oxidative stability, for example, the pasteurization process, can adversely affect both polyphenol content and antioxidant activity of beer [36,37,38]. The antioxidant activity of Witbier-style beers is enhanced by ingredients characteristic for this type of beer, that is, refined coriander and curaçao orange peel, which not only add to the taste of the finished product but also produce a strong antioxidant effect and stimulate appetite [39].

Polyphenols are bioactive substances that differ in their chemical structure, which is associated with various antioxidant and antioxidant activities and different reactions that occur during the production and storage of beer [40]. The total polyphenol content in Witbier-style beers was significant and the highest value of 156 mg/L was identified in the L25 sample. Beers made with the addition of wheat malt had considerably higher total polyphenols, on average by 17.13% compared to the beer samples made with no addition of wheat malt (Table 3). The higher contents of polyphenols in beers produced with the addition of wheat malt may be linked to the transformations in grain during the malting process and the drying process. The gradual increase in the temperature of air used in the drying process applied to malt leads to a greater intensity of changes, and to the formation of new compounds, e.g., products of Maillard reaction, as well as their isomerisation, which affects the taste (e.g., flavonol polymers) and pungency. The phenolic compounds present in beer are responsible for the sensory properties of the product. Depending on their type, the compounds may produce a sense of bitterness, pungency, richness, or contribute to the fullness of taste. Polyphenols also produce effects reflected in the intensity of beer fog [9,30]. The study by Yorke et al. [30] showed polyphenol content of 117 mg/L and 100 mg/L, respectively, in beers made with a 30% or 60% addition of unmalted wheat. Beer produced from an American wheat cultivar was found in a study by Byeon et al. [2] with total polyphenols at a level of 102.8 mg/L, whereas a study by Mascia et al. [17] reported higher values ranging between 139 and 177 mg/L, depending on the type of wheat beer. Furthermore, Depraetere et al. [5] reported total polyphenol content in wheat beers in the range of 125–142 mg/L, whereas Witbier sample investigated by Vogel [32] was found with 76.8 mg/L contents of polyphenols. The polyphenol content in beer is also related to the applied mashing process (duration and temperatures) as well as refinement of the raw materials (unmalted wheat grain, wheat malt, and barley malt) [33].

Polyphenolic compounds in Witbier-style samples were identified on an analysis of characteristic spectral data: mass-to-charge ratio *m/z* and maximum radiation absorption. Characteristics of eight polyphenolic compounds, which were found, are shown in Table 3. All identified compounds were flavonols, represented by derivatives of kaempferol, quercetin, and isorhamnetin (most frequently present in beers in glycoside form). Quercetin and kaempferol occur in combination with sugars, most commonly glucose and rhamnose, and less frequently with rutinose, arabinose, or xylose. Flavonols are antioxidant compounds that favour the cardiovascular system, they slow the aging process in human cells and inhibit the development of some cancer cells [33,41]. The contents of the investigated flavonols were significantly varied, and the highest levels of kaempferol 3-O-rhamnoside-7-O-pentoside and kaempferol 3-O-pentoside-rhamnoside were found in beer samples made from ‘Elixer’ wheat grains, compared to the remaining beer samples, whether or not produced with the addition of wheat malt (Table 3). According to Mikyška et al. [40] a slight decrease is observed in the contents of quercetin and kaempferol-O-glucoside (by 10–20% after the 6-week maturation of the beer relative to the wort) during the storage of lager-type beers. After 4 weeks of maturation of lager samples, the quercetin-O-glucoside were in the range of 0.31–0.65 mg/L, whereas the contents of kaempferol-O-glucoside ranged from 0.17 to 0.26 mg/L, depending on the wort hopping technique applied [40,42]. Kaempferol-O-glucoside and quercetins are extracted from the hops after the wort has boiled for approximately 30 min (depending on the dose) [42]. According to many researchers [26,41,43,44], mean quercetin in barley beers is in the range of 0.06 to 1.79 mg/L, and mean contents of kaempferol are at a level of 0.10 to 1.64 mg/L [41,45].

Hops, malt, and, to a lesser degree, cereal grains are a source of phenolic compounds, including flavonols [40]. The mean quercetin content in hop cones is 0.92 mg/kg d.m. and the mean content of kaempferol is 1.2 mg/kg d.m. relative to the variety and farming conditions [41,42]. Cereal-based raw materials (wheat grain, wheat malt, and barley malt) used in the production of Witbier-style beers are found with varied contents of flavanols. Suchowilska et al. [46] reported mean kaempferol in wheat grains weighing 11.4 mg/kg d.m., and quercetin at a level of 19.6 mg/kg d.m., whereas Buśko et al. [47] identified slightly lower values, amounting to 6.0 mg/kg d.m. and 6.9 mg/kg d.m., respectively. However, a study conducted by Özcan et al. [48] identified the mean contents of quercetin in barley grains at a level of 72.7 mg/kg d.m., kaempferol at a level of 19.9 mg/kg and isorhamnetin at a level of 63.5 mg/kg, whereas the malts produced from the grain had slightly higher contents of these flavonols, amounting to 81.0 mg/kg, 21.5 mg/kg and 61.9 mg/kg, respectively [48]. Worts obtained from barley malt, as a result of a mashing process, contained between 16 and 24 µg/L quercetin, depending on the barley cultivar and country of origin of the malt; however, the same wort was found without flavonol glycosides [49]. The high content of flavonol derivatives, mainly kaempferol, in the investigated Witbier-style beers may be associated with chemical transformations that take place during the process of boiling wort with hops, i.e., both with the extraction of the compounds contained in hops, and the formation of glycoside derivatives through bonding of aglycones from wheat grain and malt (quercetin, kaempferol) as well as barley malt (isorhamnetin) with sugars present in the wort, e.g., glucose, whereby O-glucoside derivatives of flavonol are formed; however, this process requires further study.

### 3.3. Sensory Analysis of Wheat Beers

The sensory qualities of the produced Witbier samples determine the specific beer style and contribute to attractiveness of the beverage for consumers. Sensory assessment of beers produced from unmalted grains of four wheat cultivars, or with the addition of wheat malt was carried out by a panel comprising 15 experts, and the results are shown in Table 4 and Table 5.

Sensory analysis showed only small differences in the ratings awarded to the Witbier samples for the general impression, except for R50, G50, and E25, and the beers were generally of good quality. The highest score in the aroma evaluation was found in the case of E50 and L50 samples. The intensity of the aroma in Witbier-style beers is mainly affected by the addition of coriander and curaçao orange peel, as well as by the quantity and quality of the hops used [50]. The taste is one of the most important qualities of beer, affecting its attractiveness and desirability for consumers. The taste and aroma profile of beer is affected not only by the raw materials used but also by certain products of the fermentation process (e.g., aldehydes, phenols and esters). The evaluation showed the highest values reflecting the taste quality in the case of G25 and R25 samples, and generally, according to the assessors, the beers enhanced with wheat malt had better taste, with the exception of the E25 sample. The remaining quality characteristics were comparable in all the investigated beers, except for the G50 sample (relatively low rating in the parameter of beer head stability) and the E25 sample (bitterness and carbonation of beer). Mild bitterness is characteristic of this beer style; furthermore, low hops beers are more acceptable for some consumers. Lutosławski et al. [50] reported similar results of sensory evaluation of Witbier-style beers [50]. The samples evaluated by the team were found with the quality characteristics reflected by the following scores: 2.90–4.53 points for aroma, 3.67–4.27 points for taste, 3.33–3.93 points for carbonation (carbon dioxide content) and 3.50–3.73 points for sense of bitterness; the evaluation was conducted on a five-point scale, in relation to the type of production type (home-made, craft and mass scale). On the other hand, the Vogel study [32] applied a nine-point hedonic scale, with responses from 1, ‘dislike extremely’, to 9, ‘extreme like’, used to evaluate aroma, taste and general impression of Witbier. The sensory characteristics were rated at 6.99, 6.76, and 6.97, respectively, on the hedonic scale, which reflects relatively good quality of the beers. The Witbier samples produced with the addition of wheat malt as a rule had more acceptable sensory qualities compared to samples made from unmalted wheat (except for samples made from grain of ‘Elixer’ cultivar) which is consistent with findings reported by de Freitas et al. [13].

Beer foam is one of the most important attributes that affects the attractiveness of beer for consumers and impacting their decision to get a given beverage again. Witbier-style beers should be characterized by ample and long-lasting frothy foam, which to a large extent is linked to the presence of glycoproteins [5]. Of all the investigated wheat beers investigated, the best quality indicators and overall impressions related to beer foam were identified in the sample G25 (Table 5). The quality and stability of beer foam is significantly affected by proteins contained in wheat grain and malts and passing into the wort during the mashing process. During the mashing process, glutenin and gliadin (gluten proteins in wheat) are transformed into proteins of high molecular mass soluble in water, whereas the malting process results in the formation of low-molecular proteins. These affect the foaming properties and hazing of beer, the effect that is particularly due to their interaction with polyphenols [5,51,52]. The proteins in the wort are stable during the boiling process and the final stage of fermentation, and the largest decrease in protein content is observed during the initial phase of fermentation. Wheat malt used in beer production positively affects the stability of beer foam, due to greater content of soluble proteins, while unmalted wheat is responsible for the size and distribution of carbon dioxide bubbles in and fine porosity of foam and, consequently, for the sensation of creamy foam [5]. A study by Wu et al. [4] showed that the fraction of proteins with a mass of 2.1–7.6 kDa was primarily responsible for beer foam, while the quality characteristics of beer are affected by the fraction of proteins with a mass of 13.2–100.0 kDa. However, we must not forget the role of proteins derived from barley malt protein-derived proteins in the quality of beer foam.

## 4. Conclusions

Adequate selection of the winter wheat cultivar for the production of Witbier-style beverages is important not only for the effectiveness of the production process and beer fermentation but also for the quality of the final product. The present study investigated the effects of a change in the ingredients defined in the recipe, in which part of the unmalted wheat was replaced with wheat malt; the findings show the best sensory profile, quality of the beer foam, and total polyphenol content in the beer samples made from material enhanced with wheat malt. Furthermore, beer samples made from unmalted wheat and barley malt (1:1) were found to have better colour, higher alcohol content, and lower energy value. These beer samples were also found to have higher antioxidant activity and flavonol content compared to the beer samples enhanced with wheat malt. Based on the findings, it may be concluded that the best were the cultivar ‘Gimantis’ (pro-healthy properties) and the cultivar ‘Lawina’ (sensory properties), and that it can be used effectively in the production of beer.

## Figures and Tables

**Table 1 foods-11-01150-t001:** Results of the physicochemical analysis of Witbier-style wheat beers.

	E50	E25	G50	G25	L50	L25	R50	R25
Apparent extract [% ;m/m]	3.87 ^cd^ ± 0.08	3.25 ^a^ ± 0.06	3.94 ^d^± 0.04	3.78 ^c^ ± 0.07	4.49 ^f^ ± 0.05	3.63 ^b^ ± 0.03	4.21 ^e^ ± 0.06	3.67 ^b^ ± 0.04
Real extract [%; m/m]	4.06 ^c^ ± 0.05	3.49 ^a^ ± 0.02	4.17 ^d^± 0.05	3.94 ^b^ ± 0.05	4.73 ^f^ ± 0.05	3.89 ^b^ ± 0.04	4.65 ^e^ ± 0.00	3.90 ^b^ ± 0.02
Original extract [%; m/m]	11.67 ^e^ ± 0.05	10.86 ^b^ ± 0.05	12.30 ^f^ ± 0.00	11.03 ^c^ ± 0.03	13.13 ^h^ ± 0.05	10.72 ^a^ ± 0.02	12.65 ^g^ ± 0.04	11.26 ^d^ ± 0.06
Degree of final apparent attenuation [%]	66.83 ^d^ ± 0.03	70.07 ^g^ ± 0.04	67.97 ^f^ ± 0.05	65.73 ^a^ ± 0.04	65.80 ^a^ ± 0.05	66.14 ^b^ ± 0.06	66.72 ^c^ ± 0.02	67.41 ^e^ ± 0.01
Degree of final real attenuation [%]	65.21 ^e^ ± 0.03	67.86 ^h^ ± 0.00	66.10 ^g^± 0.10	64.28 ^d^ ± 0.07	63.98 ^c^ ± 0.03	63.71 ^b^ ± 0.04	63.24 ^a^ ± 0.04	65.36 ^f^ ± 0.06
Content of alcohol [%; *v*/*v*]	3.92 ^d^ ± 0.03	3.78 ^c^ ± 0.06	4.20 ^e^±0.10	3.64 ^b^ ± 0.04	4.36 ^f^ ± 0.04	3.50 ^a^ ± 0.10	4.14 ^e^ ± 0.04	3.78 ^c^ ± 0.02
Content of alcohol [%; m/m]	3.10 ^d^ ± 0.03	3.00 ^c^ ± 0.00	3.34 ^e^ ± 0.04	2.89 ^b^ ± 0.03	3.47 ^f^ ± 0.03	2.78 ^a^ ± 0.00	3.29 ^e^ ± 0.01	3.00 ^c^ ± 0.05
Colour [EBC units]	6.3 ^a^ ± 0.1	7.7 ^c^ ± 0.1	6.7 ^b^ ± 0.1	7.8 ^c^ ± 0.0	6.7 ^b^ ± 0.1	8.0 ^d^ ± 0.1	6.2 ^a^ ± 0.0	8.4 ^e^ ± 0.2
Titratable acidity [0.1 M NaOH/100 mL]	2.72 ^c^ ± 0.03	2.08 ^a^ ± 0.02	2.48 ^b^ ± 0.04	2.04 ^a^ ± 0.04	2.92 ^e^ ± 0.06	2.48 ^b^ ± 0.08	2.80 ^d^ ± 0.00	2.04 ^a^ ± 0.02
pH	4.32 ^f^ ± 0.02	4.12 ^e^ ± 0.03	4.40 ^g^ ± 0.00	3.92 ^c^ ± 0.04	4.03 ^d^ ± 0.03	3.62 ^a^ ± 0.05	4.04 ^d^ ± 0.05	3.71 ^b^ ± 0.02
Bitter substances [IBU]	11.7 ^a^ ± 0.0	12.2 ^b^ ± 0.2	13.0 ^d^ ± 0.1	12.1 ^b^ ± 0.1	12.6 ^c^ ± 0.2	12.0 ^b^ ± 0.0	13.3 ^e^ ± 0.0	12.8 ^cd^ ± 0.2
Content of carbon dioxide [%]	0.43 ^a^ ± 0.03	0.43 ^a^ ± 0.00	0.44 ^ab^ ± 0.02	0.46 ^ab^ ± 0.01	0.47 ^b^ ± 0.01	0.46 ^ab^ ± 0.00	0.43 ^a^ ± 0.03	0.44 ^ab^ ± 0.04
Energy value [kcal/100 mL]	43.94 ^e^ ± 0.02	40.61 ^b^ ± 0.04	46.35 ^f^ ± 0.00	41.48 ^c^ ± 0.04	49.79 ^h^ ± 0.03	40.30 ^a^ ± 0.00	47.92 ^g^ ± 0.02	42.30 ^d^ ± 0.05

Data are expressed as mean values (n = 3) ± SD; SD—standard deviation. Mean values within rows with different letters are significantly different (*p* < 0.05). E—‘Elixer’ cultivar; G—‘Gimantis’ cultivar; L—‘Lawina’ cultivar; R—‘Rockefeller’ cultivar; 50—beer without wheat malt; 25—added wheat malt.

**Table 2 foods-11-01150-t002:** Antioxidant activity of Witbier-style beers.

	E50	E25	G50	G25	L50	L25	R50	R25
DPPH^●^ [mmol TE/L]	2.31 ^f^ ± 0.03	2.35 ^g^ ± 0.01	2.03 ^d^ ± 0.04	1.60 ^b^ ± 0.00	1.73 ^c^ ± 0.02	1.46 ^a^ ± 0.03	2.26 ^e^ ± 0.01	1.72 ^c^ ± 0.02
FRAP [mmol Fe^2+^/L]	2.12 ^e^ ± 0.06	2.08 ^de^ ± 0.03	2.09 ^de^ ± 0.01	1.90 ^b^ ± 0.03	1.74 ^a^ ± 0.04	2.01 ^c^ ± 0.03	2.05 ^cd^ ± 0.00	1.71 ^a^ ± 0.01

Data are expressed as mean values (n = 3) ± SD; SD—standard deviation. Mean values within rows with different letters are significantly different (*p* < 0.05). E—‘Elixer’ cultivar; G—‘Gimantis’ cultivar; L—‘Lawina’ cultivar; R—‘Rockefeller’ cultivar; 50—beer without wheat malt; 25—added wheat malt.

**Table 3 foods-11-01150-t003:** Polyphenol and individual phenolic compounds identified by UPLC-PDA-MS/MS in Witbier-style beers.

					E50	E25	G50	G25	L50	L25	R50	R25
Content of polyphenols [mg/L]					107 ^a^ ± 2	117 ^b^ ± 2	127 ^c^ ± 0	144 ^d^ ± 4	121 ^b^ ± 1	156 ^e^ ± 6	104 ^a^ ± 4	140 ^d^ ± 2
Compound [mg/L]	Rt [min]	λ_max_ [nm]	[M-H] m/z									
			MS	MS/MS								
Kaempferol 3-*O*-rut	3.47	276, 325	593	285	0.65 ^e^ ± 0.01	0.57 ^b^ ± 0.01	0.67 ^f^ ± 0.01	0.65 ^e^ ± 0.01	0.55 ^a^ ± 0.01	0.58 ^b^ ± 0.01	0.61^c^ ± 0.01	0.63 ^d^ ± 0.01
Isorhamnetin 3-*O*-rut	3.59	269, 325	623	315	0.49 ^a^ ± 0.01	051 ^a–c^ ± 0.02	0.50 ^ab^ ± 0.02	0.51 ^a^^–^^c^ ± 0.02	0.52 ^bc^ ± 0.02	0.53 ^c^ ± 0.02	0.51 ^a–c^ ± 0.02	0.52 ^bc^ ± 0.02
Kaempferol 3-*O*-pent-rha	3.72	272, 324	563	285	0.97 ^e^ ± 0.02	0.96 ^e^ ± 0.02	0.78 ^a^ ± 0.01	0.79 ^a^ ± 0.01	0.82 ^b^ ± 0.01	0.78 ^a^ ± 0.02	0.87 ^d^ ± 0.01	0.85 ^c^ ± 0.02
Kaempferol 3-*O*-rha	3.82	271, 324	431	285	0.65 ^a^ ± 0.03	0.81 ^d^ ± 0.03	0.66 ^ab^ ± 0.02	0.79 ^d^ ± 0.02	0.73 ^c^ ± 0.03	0.71 ^bc^ ± 0.03	0.83 ^d^ ± 0.03	0.82 ^d^ ± 0.04
Kaempferol 3-*O*-rha-7-*O*-pent	3.94	272, 324	563	431, 285	1.31 ^e^ ± 0.04	1.39 ^f^ ± 0.05	1.14 ^c^ ± 0.04	1.12 ^bc^ ± 0.04	1.04 ^ab^ ± 0.04	1.04 ^a^ ± 0.04	1.22 ^d^ ± 0.04	1.14 ^c^ ± 0.04
Quercetin 3-*O*-glc	4.57	255, 350	463	301	1.15 ^c^ ± 0.03	1.05 ^b^ ± 0.03	1.15 ^c^ ± 0.04	0.94 ^a^ ± 0.03	0.98 ^a^ ± 0.03	0.93 ^a^ ± 0.03	1.04 ^b^ ± 0.03	0.95 ^a^ ± 0.03
Kaempferol 3-*O*-glc	5.10	271, 325	447	285	0.64 ^d^ ± 0.01	0.58 ^a^ ± 0.01	0.64 ^d^ ± 0.01	0.61 ^c^ ± 0.01	0.59 ^b^ ± 0.01	0.58 ^a^ ± 0.01	0.60 ^bc^ ± 0.01	0.69 ^e^ ± 0.01
Kaempferol 3-*O*-pent-glc	5.39	271, 325	579	285	0.68 ^d^ ± 0.01	0.59 ^b^ ± 0.01	0.73 ^e^ ± 0.01	0.79 ^f^ ± 0.01	0.57 ^a^ ± 0.01	0.65 ^c^ ± 0.01	0.66 ^c^ ± 0.01	0.77 ^f^ ± 0.01
Total					6.53 ^e^ ± 0.07	6.43 ^de^ ± 0.08	6.24 ^bc^ ± 0.07	6.18 ^b^ ± 0,08	5.80 ^a^ ± 0.07	5.79 ^a^ ± 0.07	6.34 ^cd^ ± 0.08	6.34 ^cd^ ± 0.08

Data are expressed as mean values (n = 3) ± SD; SD—standard deviation. Mean values within rows with different letters are significantly different (*p* < 0.05). E—‘Elixer’ cultivar; G—‘Gimantis’ cultivar; L—‘Lawina’ cultivar; R—‘Rockefeller’ cultivar; 50—beer without wheat malt; 25—added wheat malt; rut—rutinoside; pent—pentoside; rha—rhamnoside; glc—glucoside.

**Table 4 foods-11-01150-t004:** Sensory analysis of Witbier-style beers.

	E50	E25	G50	G25	L50	L25	R50	R25
Aroma	4.07 ^a^ ± 0.58	3.46 ^a^ ± 0.43	3.80 ^a^ ± 0.67	3.80 ^a^ ± 0.67	4.00 ^a^ ± 0.36	3.73 ^a^ ± 0.22	3.60 ^a^ ± 0.52	3.87 ^a^ ± 0.19
Taste	3.93 ^ab^ ± 0.46	3.73 ^ab^ ± 0.38	3.93 ^ab^ ± 0.49	4.33 ^b^ ± 0.22	3.80 ^ab^ ± 0.21	3.87 ^ab^ ± 0.37	3.67 ^a^ ± 0.25	4.13 ^ab^ ± 0.11
Foam stability	3.73 ^ab^ ± 0.36	3.80 ^ab^ ± 0.28	3.27 ^a^ ± 0.41	4.47 ^c^ ± 0.23	3.73 ^ab^ ± 0.11	4.20 ^bc^ ± 0.46	3.73 ^ab^ ± 0.14	4.00 ^bc^ ± 0.26
Bitterness	4.13 ^a^ ± 0.15	3.44 ^a^ ± 0.26	3.88 ^a^ ± 0.50	4.13 ^a^ ± 0.28	3.81 ^a^ ± 0.28	4.06 ^a^ ± 0.33	3.81 ^a^ ± 0.32	3.94 ^a^ ± 0.31
Saturation	3.87 ^b–d^ ± 0.33	3.00 ^a^ ± 0.25	3.60 ^bc^ ± 0.28	3.40 ^ab^ ± 0.63	4.33 ^d^ ± 0.33	4.07 ^cd^ ± 0.16	3.87 ^b–d^ ± 0.18	4.07 ^cd^ ± 0.33
Overall impression	3.95 ^bc^ ± 0.16	3.49 ^a^ ± 0.31	3.70 ^ab^ ± 0.27	4.03 ^c^ ± 0.43	3.94 ^bc^ ± 0.24	3.99 ^bc^ ± 0.19	3.74 ^a–c^ ± 0.11	4.00 ^bc^ ± 0.10

Data are expressed as mean values (n = 3) ± SD; SD—standard deviation. Mean values within rows with different letters are significantly different (*p* < 0.05). E—‘Elixer’ cultivar; G—‘Gimantis’ cultivar; L—‘Lawina’ cultivar; R—‘Rockefeller’ cultivar; 50—beer without wheat malt; 25—added wheat malt.

**Table 5 foods-11-01150-t005:** Analysis of the quality characteristics of beer foam in Witbier samples.

	E50	E25	G50	G25	L50	L25	R50	R25
Colour	4.60 ^ab^ ± 0.50	4.53 ^ab^ ± 0.51	4.53 ^ab^ ± 0.33	4.80 ^b^ ± 0.41	4.33 ^a^ ± 0.17	4.47 ^ab^ ± 0.34	4.27 ^a^ ± 0.33	4.33 ^a^ ± 0.11
Abundance	4.00 ^ab^ ± 0.25	3.47 ^a^ ± 0.51	3.53 ^ab^ ± 0.23	3.73 ^ab^ ± 0.27	3.93 ^ab^ ± 0.24	4.07 ^ab^ ± 0.25	4.13 ^b^ ± 0.21	3.87 ^ab^ ± 0.09
Structure	3.33 ^a–c^ ± 0.21	3.00 ^a^ ± 0.35	3.20 ^ab^ ± 0.26	4.07 ^e^ ± 0.11	3.73 ^cd^ ± 0.11	3.67 ^b–d^ ± 0.12	3.73 ^cd^ ± 0.27	3.87 ^d^ ± 0.23
Durability	3.93 ^b^ ± 0.59	3.80 ^b^ ± 0.56	3.20 ^a^ ± 0.34	3.87 ^b^ ± 0.18	3.67 ^ab^ ± 0.17	3.93 ^b^ ± 0.23	3.80 ^b^ ± 0.25	3.80 ^b^ ± 0.13
Overall impression	3.97 ^a^ ± 0.52	3.70 ^a^ ± 0.68	3.62 ^a^ ± 0.63	4.27 ^a^ ± 0.26	3.92 ^a^ ± 0.30	4.03 ^a^ ± 0.37	3.98 ^a^ ± 0.26	3.97 ^a^ ± 0.24

Data are expressed as mean values (n = 3) ± SD; SD—standard deviation. Mean values within rows with different letters are significantly different (*p* < 0.05). E—‘Elixer’ cultivar; G—‘Gimantis’ cultivar; L—‘Lawina’ cultivar; R—‘Rockefeller’ cultivar; 50—beer without wheat malt; 25—added wheat malt.

## Data Availability

Not applicable.

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
