# Peer review of "Quality and Pro-Healthy Properties of Belgian Witbier-Style Beers Relative to the Cultivar of Winter Wheat and Raw Materials Used"

_foods, 2022, doi:10.3390/foods11081150_

Round 1

Reviewer 1 Report

The authors gave an extensive overview of the research they conducted on wheat varieties utilize for brewing. 

My remarks:

adding malted wheat and unmalted wheat is expected to give different sensory characteristics to produced beers. It is unclear why did you decide to add malted wheat to the mash if you were aiming to produce witbier?

please correct the English language

Table 1 contains commas as decimal dividers, and dots should be used, as are in the text

As I understand from the title and the abstract you aimed to determine the influence of the cultivar on the sensory and overall health profile of beers. However, you do not mention in the conclusion which cultivar was chosen best. Please rephrase the conclusion and allow the reader to learn about the cultivar. 

Also, there is a lack of information about the cultivars, no basic phisical-chemical analysis, or their properties. Why did you implement different cultivars into the research? There is no explanation why did you use different cultivars, what makes the difference between them? Is it the protein content, starch, vitreousness ...? What are the properties of chosen wheat and obtained malt?

Author Response

Authors are grateful for the contribution of the Reviewer.

adding malted wheat and unmalted wheat is expected to give different sensory characteristics to produced beers. It is unclear why did you decide to add malted wheat to the mash if you were aiming to produce witbier?

Answer:

The original Witbier beer recipe assumes the use of unmalted wheat and barley malt. The aim of the work was to be able to replace half of the the backfill of wheat grain with the wheat malt obtained from it. It was assumed that the addition of wheat malt (malting and malt drying process affect the higher share of f.e. Maillard compounds) will increase the health-promoting properties (polyphenols, antioxidant activity) of beers produced using yeast dedicated to Witbier beer and enriched with spices (coriander, bitter orange peel) and at the same time affect their sensory properties and the quality of beer foam.

please correct the English language

Answer:

The manuscript has been revised by a native speaker.

Table 1 contains commas as decimal dividers, and dots should be used, as are in the text

Answer:

It was corrected.

As I understand from the title and the abstract you aimed to determine the influence of the cultivar on the sensory and overall health profile of beers. However, you do not mention in the conclusion which cultivar was chosen best. Please rephrase the conclusion and allow the reader to learn about the cultivar. 

Answer:

Based on the findings, it may be concluded that the best was the cultivar 'Gimantis' (pro-healthy properties) and the cultivar 'Lawina' (sensory properties) and that it can be effectively used in production of beer. Lines 529-532.

Also, there is a lack of information about the cultivars, no basic phisical-chemical analysis, or their properties. Why did you implement different cultivars into the research? There is no explanation why did you use different cultivars, what makes the difference between them? Is it the protein content, starch, vitreousness ...? What are the properties of chosen wheat and obtained malt?

Answer:

All wheat used in the production of beer belong to class C (feed varieties). Wheat of the 'Elixer' cultivar is widely used in German and English malting houses and breweries. The remaining cultivars are in the stage of field research on the possibility of their use in the brewing industry. The grain crop of wheat differed in total protein content and starch content. The properties of wheat grain and wheat malts obtained from them are presented in brackets in the subsection. Lines 97-115.

Reviewer 2 Report

The authors  have performed an adequate selection of winter wheat cultivar for production of Witbier-style beverages. This is important not only for the effectiveness of production process and beer fermentation but also for the quality of the final product.

There is great work done to properly investigate the effects of a change in the ingredients defined in the recipe, whereby part of unmalted wheat was replaced with wheat malt. Interestingly, the beer samples made from unmalted wheat and barley malt (1:1) were found with better color, higher alcohol contents, and lower energy value. 

Author Response

Authors are grateful for the contribution of the Reviewer.

Reviewer 3 Report

Line 53: I understood how antioxidants are sensitive to your other factors light, oxygen, etc. Further clarification of how yeast impacts that would be nice. 

It would be nice to have a table showing the different grain builds. *By no means is it necessary, but it could help the reader have an idea what your experimental grain builds are.

Line 237 - 239: What is the standard value? 

Table 3: Could you make is so the compound names are easier to read instead of wrapping on multiple lines.

Author Response

Authors are grateful for the contribution of the Reviewer.

The manuscript has been revised by a native speaker.

Line 53: I understood how antioxidants are sensitive to your other factors light, oxygen, etc. Further clarification of how yeast impacts that would be nice. 

Answer:

Stress factors occurring during fermentation, e.g. too high temperature, too high concentration of ethanol, deficiency of nutrients for yeast, have a negative impact on yeast cells, which, defending themselves, produce reactive oxygen species damaging their structure. Reactive compounds pass to fermenting wort, in which they combine, among others, with elements such as iron or copper and form hydroxyl radicals, which contribute to the damage of compounds with antioxidant activity affecting the taste of beer. Lines 54-59.

It would be nice to have a table showing the different grain builds. *By no means is it necessary, but it could help the reader have an idea what your experimental grain builds are.

Answer:

Supplemented in the subsection material. Lines 97-115.

Line 237 - 239: What is the standard value? 

Answer:

Witbier beers are characterized by a standard extract value in the range of 11 – 13 % m/m. The recipe assumptions assumed the production of beers characterized by an extract of 11% m/m. 

Table 3: Could you make is so the compound names are easier to read instead of wrapping on multiple lines.

Answer:

Chemical abbreviations are used in table 3